# Surface Acoustic Wave Resonators for Wireless Sensor Network Applications in the 433.92 MHz ISM Band

**DOI:** 10.3390/s20154294

**Published:** 2020-07-31

**Authors:** Evangelos Moutoulas, Muhammad Hamidullah, Themis Prodromakis

**Affiliations:** Centre for Electronics Frontiers, Zepler Institute for Photonics and Nanoelectronics, University of Southampton, Highfield Campus, University Road, Building 53 (Mountbatten), Southampton SO17 1BJ, UK; M.Hamidullah@soton.ac.uk (M.H.); T.Prodromakis@soton.ac.uk (T.P.)

**Keywords:** metallization ratio, metal thickness, ISM band, process control, resonator, surface acoustic waves, WSN, resonator arrays

## Abstract

Surface acoustic wave (SAW) resonators are low cost devices that can operate wirelessly on a received radio frequency (RF) signal with no requirement for an additional power source. Multiple SAW resonators operating as transponders that form a wireless sensor network (WSN), often need to operate at tightly spaced, different frequencies inside the industrial, scientific and medical (ISM) bands. This requires nanometer precision in the design and fabrication processes. Here, we present results demonstrating a reliable and repeatable fabrication process that yields at least four arrays on a single 4-inch wafer. Each array consists of four single-port resonators with center frequencies allocated inside four different sub-bands that have less than 50 kHz bandwidth and quality factors exceeding 8000. We see promise of standard, low-cost photolithography techniques being used to fabricate multiple SAW resonators with different center resonances all inside the 433.05 MHz–434.79 MHz ISM band and a mere 100 kHz spacing. We achieved that by leveraging the intrinsic process variation of photolithography and the impact of the metallization ratio and metal thickness in rendering distinct resonant frequencies.

## 1. Introduction

Wireless sensor networks (WSNs) consist of spatially distributed sensors that record the physical conditions of the environment. Especially in industry, they are increasingly being deployed to assist the maintenance of industrial equipment and infrastructure. WSNs that employ passive surface acoustic wave (SAW) sensors are being investigated [1,2] due to the commercial availability of such devices. These networks work on radio frequency signals and can operate either in the time or the frequency domain, in order to distinguish the signals received by the individual sensors on the network. While research focuses on network architectures [3] and radio frequency interrogation strategies [4], a key feature for the frequency domain interrogation strategy is having sensors with distinct frequency responses that also operate within specified frequency bands, due to industrial restrictions. However, precise fabrication of SAW devices is required due to the relatively narrow range of these bands.

The fabrication of SAW resonators of different materials and designs has been studied extensively, particularly for use as filters [5], sensors [6,7] and RFID tags [8]. In fact, most of the manufacturing challenges have been addressed, since those devices found their way into industry twenty years ago [9,10]. Ongoing research is focused on new materials and applications [11]. In industrial applications wherein sensor networks are deployed, these devices are especially promising candidates due to their low cost, robustness and wireless operation. Although obtaining resonators with a desired frequency is not a problem with modern fabrication tools [12,13], fabrication process complexity and the large number of factors that affect their performance pose a challenge when aiming for an array of devices with similar, but distinct frequency responses; simply put, getting a number of SAW resonators to operate inside the 433.05–434.79 MHz industrial, scientific and medical (or 433.92 MHz ISM) band, with unequal but tightly spaced resonance frequencies and high quality factors, is challenging due to the susceptibility of such electrical characteristics to intrinsic fabrication process variations [14,15]. As a result, most industrial SAW resonators require further individual plasma treatment after cleanroom fabrication, so that the resonance is tuned to the targeted frequency. This is especially challenging for differential SAW resonators with different orientations, since etching will not affect the two resonators in the same way. In practice, when wireless operation is desired and a reader module is used to wirelessly interrogate the resonators in the frequency domain, the reader’s accuracy will dictate the frequency separation that is needed between the distinct responses and subsequently the number of resonators in a multi-resonator arrangement [16] operating in the 1.8 MHz range of the 433.92 MHz ISM band.

The capability of producing single SAW resonators with resonant frequency inside the 433.92 MHz ISM band has been proven in many studies and for different purposes. Varshney et al. [17] investigated the transient response of a commercially available resonator. Forsen et al. [18] used nano-imprint lithography (NIL) to fabricate resonators in a wide range of frequencies, with performance and yield comparable to industrial processes. Kalinin et al. [7] demonstrated torque sensors based on SAW resonators with frequencies in the 433–437 MHz range. At the same time, a number of systems and networks have been proposed in a wide range of operating frequencies [19,20,21], indicating the increased interest in such applications. However, none of them is focused on achieving operation inside the mentioned frequency band due to the difficulties in fabricating several resonators with distinct frequencies in such a small range. Cunha et al. [22] reported several langasite devices, each based on a different design with different target resonant frequencies in the 2.45 GHz ISM band. They were fabricated on the same wafer using electron beam (EBL) and ion milling lithography. The bandwidth of this band (100 MHz), however, enables a 10 MHz resonant frequency difference between their devices, which is ten times wider than the entire range of the 433.92 MHz band. Therefore, the error margin during fabrication is considerably larger than in our case. Bruckner and Bardog [23] recently demonstrated the wireless interrogation of four temperature sensors based on delay lines in the same 2.45 GHz ISM band, but those devices were designed for time domain interrogation wherein there is no need for frequency separation.

In this study, we attempted to fabricate high performance SAW resonators on the well-studied ST-cut quartz [24,25], while quantifying the process variation of our tools, in order to utilize it for obtaining several devices with separable frequency responses within the 433.92 MHz ISM band. This was done by examining the susceptibility of three characteristics of the interdigitated transducer (IDT) to process variations and their effects on the device performance. Those were the wavelength (l), the metallization ratio (M) and the metal electrode thickness (t). Their effect was simulated and then evaluated after the fabrication. We evaluated the characteristics of 24 resonators produced by the same design and uniformly distributed on the same wafer. The statistical analysis allowed us to indirectly “control” the resonant frequencies fitted in a very narrow range. Finally, we used the results to create a fabrication approach that provides several SAW resonators which cover the entire 433.05–434.79 MHz ISM band and can be fabricated with standard micro-fabrication tools.

## 2. Materials and Methods

### 2.1. Fabrication Process

The following procedure was followed during fabrication. Prior to UV exposure, the wafer was dehydrated in an oven at 200 °C, for 30 min; exposed to HMDS adhesion promoter vapors at room temperature, for 5 min; and then baked on a hotplate at 110 °C, for 60 s. AZ nLOF 2020 negative photoresist was spun at 2500 rpm, producing a 2 μm thick resist layer; that was followed by a soft hotplate bake at 110 °C, for 60 s. The resist was then patterned using a mask in hard contact with the wafer and exposure under UV light. It was found that the optimal dose was i-line at 72 mJ/cm^2^. Hard bake on the hotplate, at 110 °C, for 60 s, was used after that to stabilize the resist. Finally, the wafer was immersed by hand in AZ 726 MIF developer, for 20 s, at room temperature, and immediately immersed in de-ionized water.

Deposition of the Ti (10 nm) and Al (150 nm) layers was done using a Leybold Lab 700 e-gun evaporator. This technique was chosen over other deposition techniques, like sputtering, since it produces less conformal metal coating that assists the effective lift-off. The Ti layer greatly improved the adhesion of the deposited metals on the substrate, and it had no significant result on device performance. Lift-off was done in NI55 high performance stripper bath for more than 24 h. The high performance stripper left no metal residue, so sonication was not required.

### 2.2. Electrical Characterization

The fabricated devices were electrically characterized using a Cascade probe station and a ZNB40 Rhode Swartz vector network analyzer (VNA). All the measurements presented are wired measurements with typical coaxial cables and RF probes calibrated in the examined frequency range. Using these measurements the resonant frequencies were determined and the quality of our design was verified by calculating the bandwidth and the Q-factor.

## 3. Results

### 3.1. Design and Simulation

The most common type of SAW device employs Rayleigh waves (RSAW), where the coupling of longitudinal and shear waves confines the acoustic energy near the surface of the substrate creating an out-of-plane elliptically polarized surface wave [26]. The frequency (*f*) of such waves is calculated the following equation:(1)f=v/l
where *v* is the velocity of the wave for the given substrate material, and *l* is the wavelength. However, the actual frequency response of a SAW device will be affected by the mass loading, as described by the perturbation theory [27]. As a result, the added mass of the metal that makes up the IDT will significantly impact the result but at the same time it will provide the means to control the frequency by controlling the geometric characteristics of the IDT, meaning the metallization ratio, which is defined as the finger width divided by the pitch (l/2), and the metal thickness. Higher metal thickness and metallization ratio will increase the mass loading effect and thus reduce the resonant frequency. Therefore, simulations are necessary when one tries to design and fabricate SAW devices with accuracy.

Using COMSOL Multiphysics, RSAW 3D primitive cell simulations were performed for different wavelengths from 7.180 μm to 7.220 μm. As shown in Figure 1a, the operating frequency is inversely linear to the wavelength. To obtain the operating frequency within the ISM radio band, the minimum and maximum wavelengths are 7.1825 μm to 7.2145 μm respectively. The 32 μm margin between minimum and maximum wavelength demands a precisely designed and well controlled fabrication process variation. Furthermore, the requirement is even higher if several resonators are required within the ISM band at different operating frequency. For instance, to get 5 resonators within the ISM band with around 300 kHz center frequency spacing between them, the difference between each resonator wavelength must be equal to 6 μm. For an IDT design with 0.667 metallization ratio for instance, the IDT finger widths for 5 resonators are 2.396 μm, 2.398 μm, 2.400 μm, 2.402 μm and 2.404 μm. It is very difficult to perform the electrode patterning process, even with the most advanced EBL [28], because it will require an EBL spot size of 2 μm or lower.

In addition to the wavelength, the metal electrode design will also affect the resonant frequency due to the mass loading effect, as mentioned before. Figure 1b shows how the resonant frequency is affected by the metal thickness and the metallization ratio. It can be seen that the resonant frequencies are lower for higher metal thicknesses and metallization ratios, as expected. For example, based on the simulation, a ±10 nm variation in the metal thickness will result in a ±400 kHz variation in the resonant frequency (for a metallization ratio of 0.65). These results suggest that there is relatively wider variation margin in metal thickness and metallization ratio than the wavelength variations. Thus, by utilizing and controlling wafer level process variation, several resonators working at different resonant frequencies within 433.05–434.79 MHz ISM band can be obtained.

Acquiring a SAW resonator with high Q-factor depends on the IDT and reflector design, which must have 50 Ohm real impedance at the resonant frequency, where the imaginary impedance is zero. That way the maximum power will be transferred from the antenna/cable to the resonator. The number of IDT fingers, the aperture and the metallization ratio will affect the impedance value. Furthermore, the resonator performance can be improved by adding several additional features such as quasiconstant acoustic reflection periodicity (QARP) and apodization structure [29] to suppress the bulk-scattering loss and minimize parasitic peaks. In this work, we fabricated resonators using a design based on proven standards [6], with Al deposited on ST-cut quartz substrate. The IDT and the reflector gratings (Figure 2b) have the same wavelength, and the IDT also features an apodization pattern. The SAW resonator is designed to have a resonant frequency of 434 MHz. The number of Al strips in the gratings is 275, the number of fingers in the apodized IDT is 140, and the aperture is 0.37 mm. Based on the SAW Resonator COM model [30], the 50-Ohm impedance match is obtained at metallization ratio of 0.65, equivalent to electrode width and spacing of 2.34 and 1.26 μm respectively. These dimensions introduce a complexity to the electrode patterning process, especially because the gratings and IDT finger spacing are close to the limit of standard photolithography. Figure 2c shows the effects of this limitation on the geometrical characteristics of the fabricated devices around the apodization features. The actual dimensions were subject to intrinsic variations but the device remained functional. Thus, the choices of photoresist materials, photolithography techniques and metal deposition process are highly critical to producing a reliable and repeatable result.

### 3.2. Saw Resonators on ST-Cut Quartz

The sequence of standard cleanroom processes that were followed during fabrication are summarized in Figure 2a. A 4-inch ST-cut quartz wafer along with standard photoresist patterning, Al deposition via sputtering and the lift-off process were chosen, thereby keeping the cost low, compered to other methods, such as EBL. Throughout fabrication, each step was evaluated and the next section shows how these findings can be used as a means to obtain resonators with distinct frequency responses, from the same design and on the same wafer.

These standard methods were optimized for the selected materials and design. For glass substrates such as ST-cut quartz, photoresist adhesion can be a problem and an adhesion promoter was used to overcome this. Transparency of the substrate is also an issue when it comes to resist patterning with UV light. The lack of reflections at the bottom of the resist can affect the resist profile but most importantly for structures like the 1.26 μm gaps in our design, diffraction phenomena can create electrical short circuits between the Al stripes of the IDT and cause failure of the device. Therefore, careful selection of the exposure dose and the use of hard contact between the mask and the resist were important. Finally, poor adhesion of the Al thin film on the glass substrate required the addition of a thin Ti adhesion layer.

### 3.3. Evaluation of Process Variations

The metallization ratio and the metal electrode thickness were measured during fabrication to evaluate the intrinsic process variation of those two parameters. The first was measured using an optical microscope, and the average value for each resonator is presented. The second was measured with a stylus profilometer for each resonator as well.

The evaluation of the metallization ratio is displayed in Figure 3a with the parameter ranging from 0.4 to 0.7. The mean value was 0.571 with 0.083 standard deviation, while the target value from the design was 0.65. The variation around the mean value is associated with the photolithography step and is due to the fundamental limitations that arise from both the exposure to UV light and the manufacturing of the photo-mask itself. For example, rounding errors between the design and the mask fabrication tools may create variation across the large number of the consecutive stripes that form the IDT and reflector. But even for a mask with finer features, light diffraction would cause variations in the width of the stripes across the wafer.

Similarly, the variation of the metal thickness on the wafer is displayed in Figure 3b. The mean value for the metal thickness was found to be 153.2 nm and the standard deviation 2.5. There is a significant difference of 6.8 nm between the mean and the target value of 160 nm. This systematic error could appear due to deposition, which produced a thinner film than the target. It could also appear due to inaccurate thickness measurements with the stylus profilometer.

Although those variations would normally be an undesired effect, in this case they can be utilized to introduce a dispersion of the resonator characteristics, around the targeted values, which will result in similar but different resonant frequencies. Despite the lower than expected metallization ratio and the metal thickness, according to the simulation, most of the devices will have resonant frequencies inside the ISM band (Figure 2b). For the measured mean values presented above, the mean resonant frequency, according to the simulation, was expected to be 434.5 MHz. The electrical characterization results show that the mean frequency obtained was lower than that, as will be discussed in the next sections.

### 3.4. Saw Resonator Performance

The measured S11 scattering parameter response shown in Figure 4 presents the comparison between two fabricated resonators with the same design characteristics, but one of them featured an apodized IDT. The two resonators had different resonant frequencies; therefore, the x-axis was normalized so that a comparison between the two responses was possible. The result shows that this technique suppresses parasitic peaks around the resonant frequency. This can reduce interference between signals in multi-resonator arrays.

The real and imaginary admittance plots can be found in Figure 5. The resonant frequency for each resonator was extracted from the imaginary admittance plot. The imaginary admittance will be zero at the resonant and anti-resonant frequencies. The bandwidth was extracted from the real admittance plot and corresponds to the FWHM of the peak. The Q-factor here is defined as the resonant frequency divided by the calculated bandwidth. By converting the admittance (*Y*) to impedance (*Z*) using the equation:(2)Z=1/Y
we can determine the real impedance value at the resonant frequency. This is also the point whereat the imaginary impedance will be zero; therefore, the system will act as a closed circuit. For the resonator that corresponds to Figure 5, the real impedance at the resonant frequency is 52 Ohm. This value is very close to the 50 Ohm target but a matching circuit can be used if the reflected power from this mismatch is significant.

### 3.5. Variation Across a Single Wafer

A wafer with 24 resonators uniformly distributed on it was fabricated. Figure 6a–c shows the variations of the resonant frequency (MHz), the bandwidth (Hz) and the Q-factor across the devices. After the characterization, 22/24 resonators were fully functional, resulting in a 91.6% yield. The distributions presented are a result of the distributions of the metallization ratio and metal thickness presented in Figure 3a,b, respectively. Most importantly, the combination of those two distributions results in a ±160 kHz variation in the resonant frequency, as seen in Figure 6a. A resonator designed for interrogation in the frequency domain must have high quality factor and small bandwidth, so that a reader module can distinguish each resonant frequency without any interference between the responses. In our case, the maximum bandwidth was 54 kHz and most of the devices had a bandwidth around 49 kHz, as seen in Figure 6b. Moreover, the Q-factor was between 8000 and 9800, as demonstrated in Figure 6c. These values are sufficient for a standard reader module to be able to effectively interrogate the resonators wirelessly [16].

Statistical analysis was performed on the data from Figure 6 to determine whether the resonant frequency follows a normal distribution around the mean and whether the mean frequency of the resonators produced with our process is significantly different from the expected frequency, calculated in the simulation. The mean value for the distribution of the resonant frequencies was 433.913 MHz and the standard deviation 0.149. According to the Shapiro–Wilk test for normality the data do indeed follow a normal distribution with a *p*-value of 0.113. This test was chosen because it is more suitable for small size samples. With this assumption, a one-sample t-test was performed but the mean value was found to be significantly different from the expected 434.5 MHz, at the 0.05 level. During the evaluation of the process variation, the average metallization and the average metal thickness were found to be lower than the targeted values from the design.

## 4. Discussion

All of the functional resonators have a resonant frequency inside the ISM band. By combining this analysis with the metallization ratio measurements and the metal thickness measurements we can conclude that the intrinsic variation of those two parameters is responsible for the variation of the resonant frequency. Therefore, they can be used in a systematic manner to produce several resonators with distinct frequencies, within a small frequency range.

As discussed before, according to the simulation, the measured metallization ratio and metal thickness were expected to result in a higher mean frequency (434.5 MHz) than the targeted (434 MHz). However, the mean measured frequency (433.913 MHz) was found 600 kHz lower. This error is less than 0.15%. Possible reasons for this difference are systematic errors in the measurements, especially for the metal thickness, and inaccuracies of the simulation, which did not consider the effects of the reflector gratings and damping losses. This finding can be used to improve the accuracy of the simulation in future. Most importantly, the mean resonant frequency was only 90 kHz lower than the targeted value of 434 MHz.

The distribution of frequencies that was explored in the results section allows us to further divide the frequency range into four sub-bands, each with 150 kHz range. That means we can have resonators working in each sub-band with at least 100 kHz separation between their frequencies. This enables the simultaneous interrogation of four sensors while we are able to clearly distinguish their frequency responses. Finally, by using multiple channel interrogation in the frequency domain [6], or time domain interrogation [23], we can increase the number of resonators in a wireless network even more.

Figure 7 shows the S11 response of every fabricated resonator on the same wafer and highlights the 4 × 4 arrays obtained. These arrays were formed by grouping 16 of the 24 fabricated resonators based on their resonant frequencies, in such a way that every array had four distinct resonances. Even though this configuration was chosen to demonstrate as many sets as possible, other configurations with more resonators can be chosen, depending on the application. These sets meet the criteria for use in applications with resonator arrays interrogated in the frequency domain and resonant frequencies inside the 433.92 MHz ISM band.

The mathematical analysis on the distribution of frequencies provided us statistical confidence that wafers produced using the fabrication process described in the methods section will have enough resonators with resonant frequencies in each sub-band. As a result, one will always be able to find at least four resonators in each of the four sub-bands and form the desired arrays. While the photo-mask used here included several devices intended for different research purposes and allowed for a limited number of resonators to be produced, by using a photo-mask that includes only such resonators, a significantly larger number of them can be produced to increase the number of arrays obtained. The uniform distribution of the resonators on the fabricated wafer statistically represents a wafer with only these devices on it. It is worth noticing that these results were achieved using only one design on the photo-mask and cover only a portion of the ISM band. By including three designs with resonant frequencies distributed inside the band—each following a normal distribution like the one shown in Figure 7a, but a different mean value—it was possible to expand the range of the fabricated resonators to cover the entire band. As a result of the variations described, one can potentially produce even more sets, uniformly distributed across the desired range. Furthermore, this approach can be adjusted and used for a different frequency range, if operation inside an ISM band is not required.

The capability of fabricating several high Q and narrow bandwidth SAW single-port resonators on ST-cut quartz with distinct resonant frequencies inside the 433.92 MHz ISM band was demonstrated using standard cleanroom tools. First, we investigated the impacts of two parameters (metallization ratio and metal thickness) on the resonant frequency and measured their intrinsic process variation during fabrication. It was then found theoretically and experimentally that by utilizing these variations we can overcome the nanometer precision required to effectively produce several resonances using a statistical approach. This approach allows us to indirectly “control” the resonances achieved on a wafer, using only one standard design. We were able to produce 4 × 4 sets of resonators with the mentioned characteristics, all fabricated on the same wafer. The proposed process is low-low cost, suitable for a glass substrate and suitable for mass manufacturing. It can be used to produce resonators for applications wherein a small resonant frequency separation between devices is desired. Such applications include all WSNs that involve resonators, but especially ones deployed in industrial and harsh environments.

## Figures and Tables

**Figure 1 sensors-20-04294-f001:**
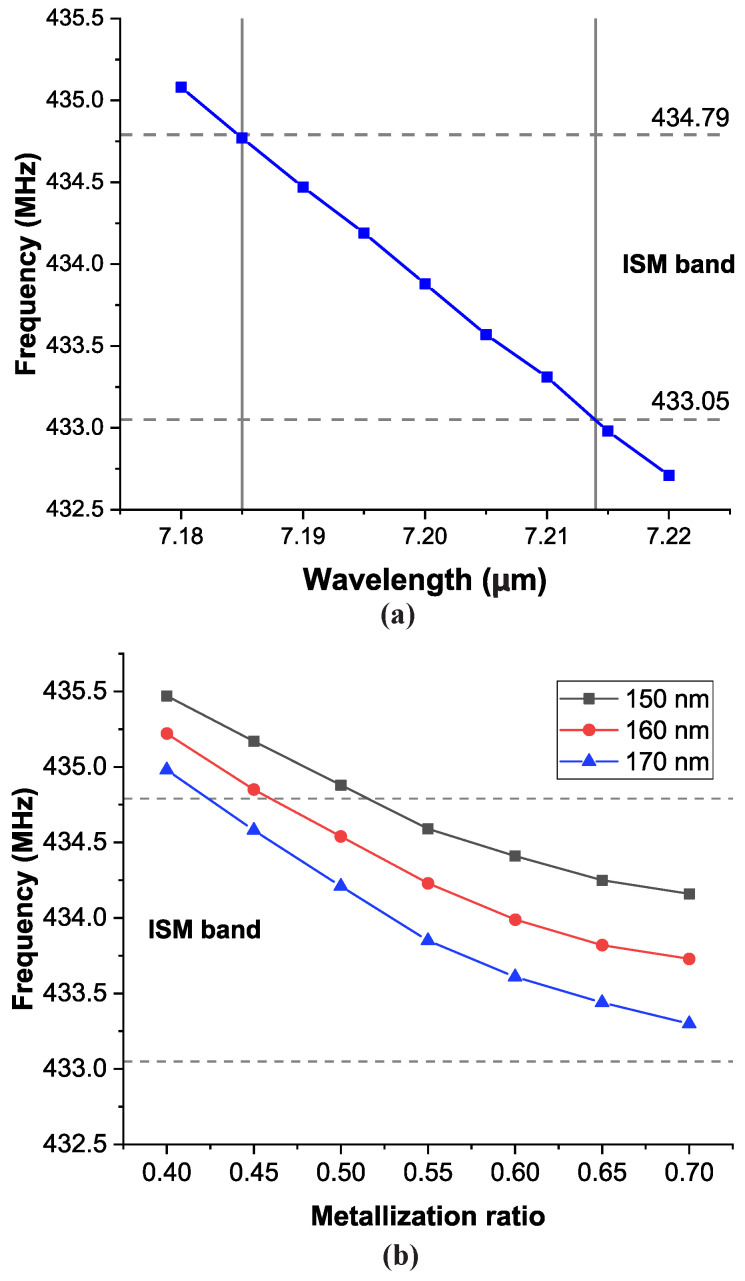
COMSOL Multiphysics simulation of lST-cut quartz SAW resonator 3D primitive cell. (**a**) Eigenfrequency vs. Interdigittated transducer wavelength at 160 nm Al metal thickness and 0.667 metallization ratio and (**b**) eigenfrequency vs. metal thickness for three different metal thicknesses and 7.2 μm wavelength.

**Figure 2 sensors-20-04294-f002:**
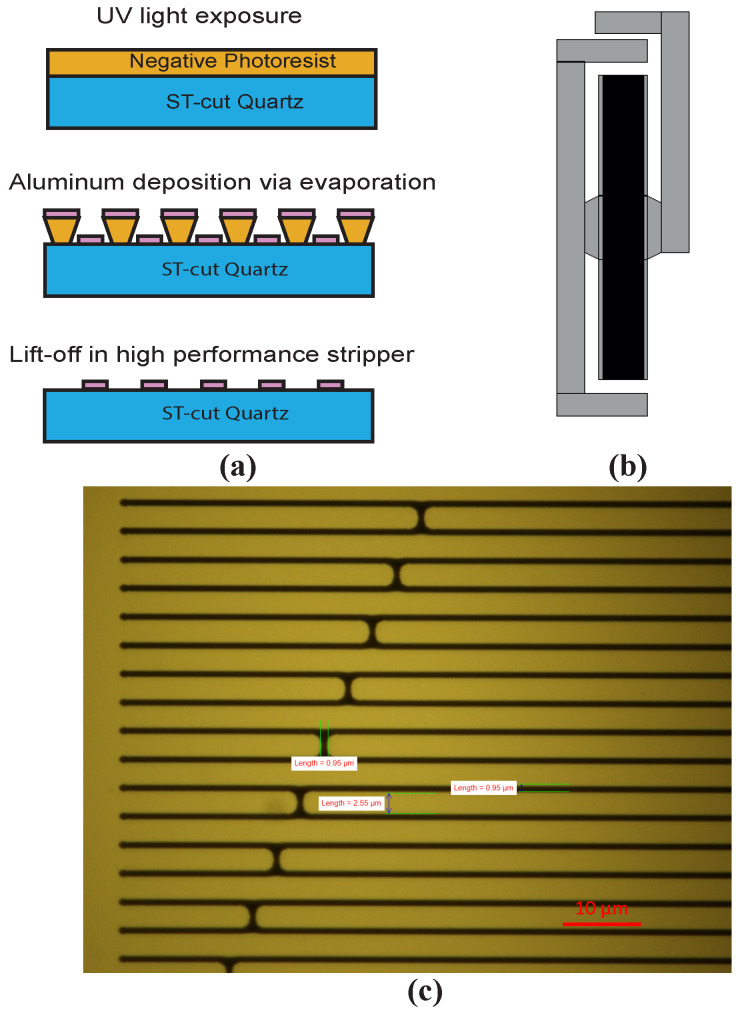
Resonator fabrication: (**a**) main process steps; (**b**) fabricated device featuring IDT, reflector gratings and electrode pads for electrical characterization; (**c**) IDT apodization features and critical dimensions.

**Figure 3 sensors-20-04294-f003:**
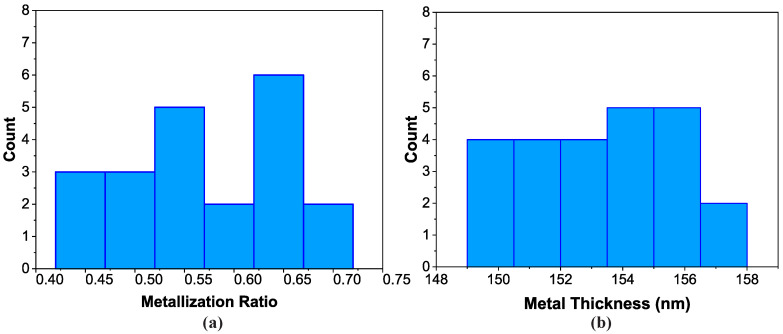
Evaluation of process variation across a single wafer bearing 24 resonators: (**a**) distribution of the metallization ratio measurements using an optical microscope, (**b**) distribution of metal thickness measurements using a stylus profilometer.

**Figure 4 sensors-20-04294-f004:**
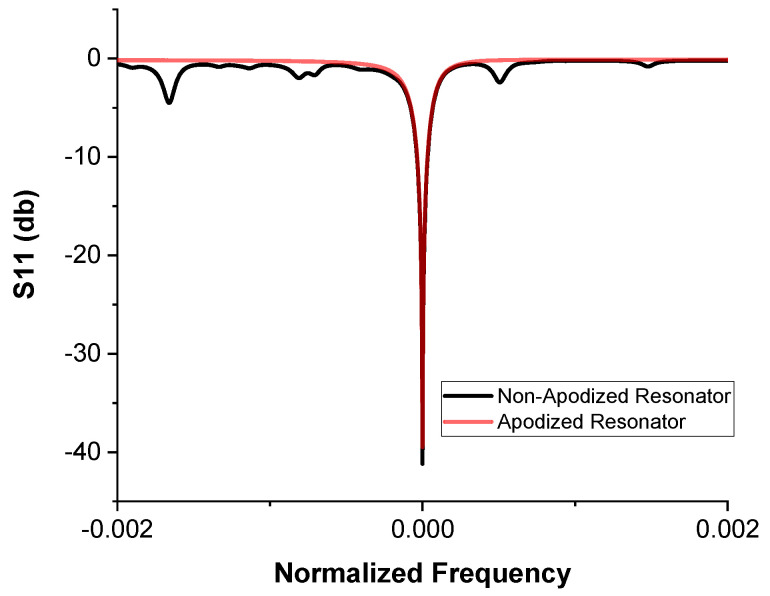
The effect of apodization. S11 responses for apodized and non-apodized resonators.

**Figure 5 sensors-20-04294-f005:**
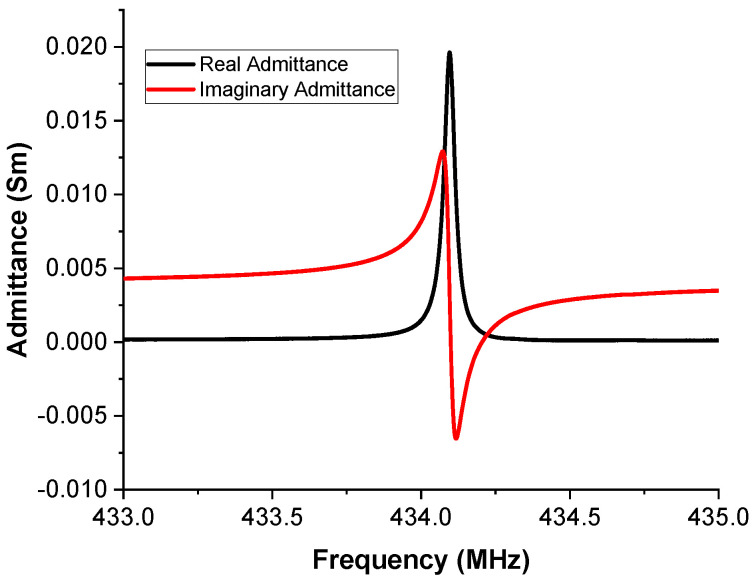
Resonator performance. Admittance measurements for single resonator using a vector network analyzer.

**Figure 6 sensors-20-04294-f006:**
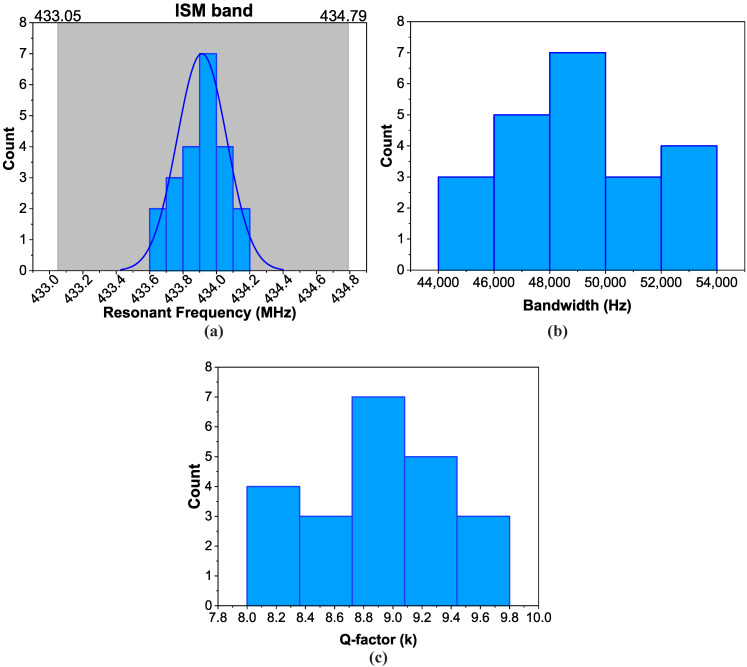
Distribution of electrical characteristics for 24 resonators on the same wafer: (**a**) resonant frequency extracted from the S11 response, (**b**) bandwidth extracted from the admittance response, (**c**) calculated Q-factor.

**Figure 7 sensors-20-04294-f007:**
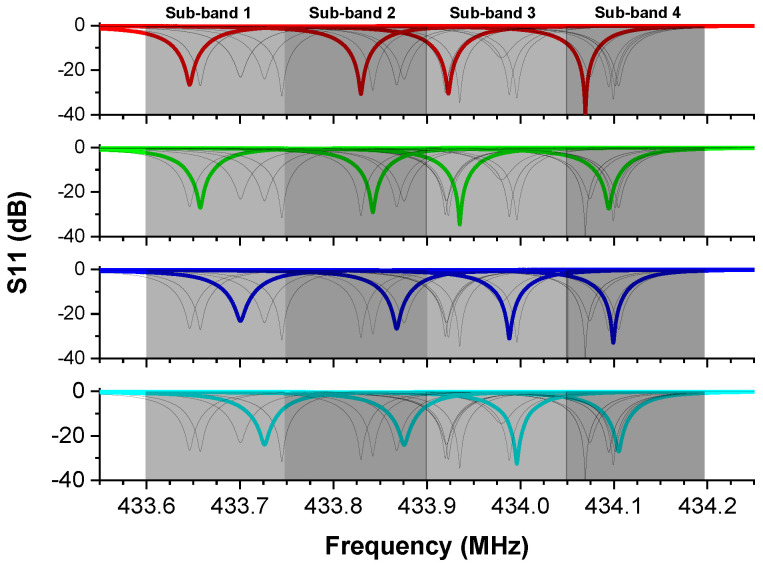
S11 response of every resonator on a single wafer. 4 × 4 sets of SAW resonators can be formed with at least 100 KHz frequency separation for each set.

## Data Availability

All data supporting this study are openly available from the University of Southampton repository at https://doi.org/10.5258/SOTON/D1491.

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
