# Peer review of "Surface Acoustic Wave Resonators for Wireless Sensor Network Applications in the 433.92 MHz ISM Band"

_sensors, 2020, doi:10.3390/s20154294_

Round 1

Reviewer 1 Report

The authors tackle the well known issue of SAW resonator cleanroom fabrication reproducibility in order to try and fit up to five resonances in the European ISM band.

page 3: the uncertainty analysis involves electrode period (wavelength, Fig 1a) and a/p (Fig 1b) but is missing the core issue of metalization thickness which is the source of inhomogeneity across a wafer due to varying mass loading impact.

page 6: I was very concerned when I read "to  calculate the resonant frequency for each resonator. It corresponds to the S11 minimum" which is incorrect. Fortunately the authors correct their mistake in the following page with the Q factor calculated on the width at half height of the real part of the admittance. Yet  how the "real part of the impedance that corresponds to the resonant frequency" is closer to the truth the yet not correct (how can a real part of Z be a frequency ?)

page 8: the authors claim the need for nm resolution lithography, but end up with the usual distribution of resonance frequencies of about 200 kHz at 434 MHz, with a statistics on only 24 devices (why so few on a 4" wafer ? we have about 1000 resonators on a 4" wafer produced by SENSeOR). No strategy for selecting a subset of the devices (sorting) nor differential measurement -- mandatory to get rid of the fluctuations of the reference oscillator when probing the passive sensor response -- which allows for pairing devices with similar frequency offsets is addressed.

The issue of lithography resolution using novel patterning techniques -- beyond ebeam lithography which the authors correctly as hardly usable in an industrial context -- has been for example tackled in https://ieeexplore.ieee.org/document/5936002/ which the authors might be interested in referring to in future investigations.

Minor detail: the authors claim the need for the resonator to exhibit 50 ohm impedance at resonance. This is not correct in a wireless measurement configuration in which the only impedance matching condition is for the resonator to exhibit an impedance which is the complex conjugate of the antenna impedance at the operating frequency. Since antennas hardly ever exhibit 50 ohms, and actually vary quite strongly with environment, the SAW resonator might be adapted to these operating conditions.

I cannot see anything technically wrong in the paper, but what is the novelty ? SAW resonators have been designed and produced for decades with better performances and yield than those shown in this paper, so I cannot figure out what result is worth publishing.

Reviewer 2 Report

The article is devoted to the rather complicated technical problem of creating the several surface acoustic wave resonators on a single quartz plate with the given parameters. The authors performed calculations of such resonators and have found not only the optimal values of all geometrical sizes of interdigital transducers and reflectors but also their optimal metal thicknesses and metallization ratios. The article can be published with minimal corrections in accordance with the following:

  1. The number of resonators on one plate remains unclear. In abstract (lines 5-9) authors inform that the structure contains 4 arrays with 4 resonators. So the number of resonators is equal 16. On the line 171 the authors write “A wafer with 24 resonators on it was fabricated.” The number of resonators should be clearly indicated.
  2. For a better understanding of figure 2, it would be good to add to the Figure 2 a quartz plate with located resonators, including transducers and reflectors.
  3. SAW resonators are single- and double-port. The authors do not specify the type of resonators and it will be good to indicate that single-port resonators are being developed.
  4. It is necessary to explain in more detail how to understand figures 3 and 6.

Reviewer 3 Report

The work is interesting and well presented. My opinion is that it has to be published.

Author Response

Thank you for your reply.

Round 2

Reviewer 1 Report

The authors have updated the manuscript by answering all comments from this reviewer and correcting the various uncertainties as to their analysis. While the results are well known from the SAW manufacturing industry, I have not found published references reporting such results and have to admit that the paper is worth publishing. I would nevertheless mention that most industrial surface acoustic wave resonators require individual trimming after cleanroom manufacturing to tune the resonance to the targeted frequency (a challenge for differential SAW resonators since both resonator orientations are not affected the same way during the plasma etching trimming procedure). Furthermore, I would provide reference to some of the older literature to emphasize that SAW resonators have been used for a long time, with all the associated manufacturing challenges met by Siemens, Vectron or SAW Components to only name the largest providers of SAW transducers for passive wireless sensing. Examples of such references include

W. Buff, F. Plath, 0. Schmeckebier, M. Rusko, T. Vandahl,H. Luck, F. M ̈oller, and D.C. Malocha. Remote sensor systemusing passive saw sensors. InIEEE Ultrasonics Symposium,pages 585–588, 1994.

A.Pohll, F.Seifert, L.Reind, G.Scholl, T. Ostertag, and W.Pietschl. Radio signals for saw id tags and sensors in strong electromagnetic interference. In IEEE Ultrasonics Symposium,pages 195–198, 1994.

L. Reindl, G. Scholl, T. Ostertag, C.C.W. Ruppel, W.-E. Bulst,and F. Seifert. SAW devices as wireless passive sensors. In IEEE Ultrasonics Symposium, pages 363–367, 1996.

W. Buff, M. Rusko, M. Goroll, J. Ehrenpfordt, and T. Vandahl. Universal pressure and temperature SAW sensor for wireless applications. In IEEE Ultrasonics Symposium, pages 359–362,1997.

A. Pohl and L. Reindl. Measurement of physical parameters of car tires using passive saw sensors. In Advanced Microsystems for Automotive Applications Conference, pages 250–262, 1998.

W. Buff, S. Klett, M. Rusko, J. Ehrenpfordt, and M. Goroli. Passive remote sensing for temperature and pressure using SAW resonator devices.IEEE Transactions on Ultrasonics, Ferroelectrics, and Frequency Control, 45(5) :1388–1392, september 1998.

V. Kalinin. Influence of receiver noise properties on resolution of passive wireless resonant saw sensors. In IEEE Ultrasonics Symposium, volume 3, pages 1452–1455, 2005

although Kalinin's 2006 conference paper is already cited and might be sufficient.
